# Field evaluation of the performance of seven Antigen Rapid diagnostic tests for the diagnosis of SARs-CoV-2 virus infection in Uganda

**Josephine Bwogi**[1]*, **Tom Lutalo**[1], **Phionah Tushabe**[1], **Henry Bukenya**[1], **James Peter Eliku**[1], **Isaac Ssewanyana**[2], **Susan Nabadda**[2], **Christopher Nsereko**[3], **Matthew Cotten**[4], **Robert Downing**[1], **Julius Lutwama**[1], **Pontiano Kaleebu**[1,4], **EPI Laboratory team**[¶], **UVRI -COVID 19 Technical team**[¶]

1 Uganda Virus Research Institute, Entebbe, Uganda, 2 Central Public Health Laboratories, Kampala, Uganda, 3 Entebbe Regional Referral Hospital, Entebbe, Uganda, 4 Medical Research Council/Uganda Virus Research Institute & London School of Hygiene and Tropical Medicine, Uganda Research Unit, Entebbe, Uganda

☯ These authors contributed equally to this work.
¶ Membership of the EPI Laboratory team and Membership of the UVRI COVID-19 Technical Team are provided in the Acknowledgments.
* josephinebwn12@gmail.com

**Data Availability Statement:** The data underlying the results presented in the study are available from: http://eaccr.org/sites/default/files/2021-11/

## Abstract

### Objective

The objective of this study was to evaluate the performance of seven antigen rapid diagnostic tests (Ag RDTs) in a clinical setting to identify those that could be recommended for use in the diagnosis of SARS-CoV-2 infection in Uganda.

### Methods

This was a cross-sectional prospective study. Nasopharyngeal swabs were collected consecutively from COVID-19 PCR positive and COVID-19 PCR negative participants at isolation centers and points of entry, and tested with the SARS-CoV-2 Ag RDTs. Test sensitivity and specificity were generated by comparing results against qRT-PCR results (Berlin Protocol) at a cycle threshold (Ct) cut-off of ≤39. Sensitivity was also calculated at Ct cut-offs ≤29 and ≤33.

### Results

None of the Ag RDTs had a sensitivity of ≥80% at Ct cut-off values ≤33 and ≤39. Two kits, Panbio™ COVID-19 Ag and VivaDiag™ SARS-CoV-2 Ag had a sensitivity of ≥80% at a Ct cut-off value of ≤29. Four kits: BIOCREDIT COVID -19 Ag, COVID-19 Ag Respi-Strip, MEDsan® SARS-CoV-2 Antigen Rapid Test and Panbio™ COVID-19 Ag Rapid Test had a specificity of ≥97%.

Field%20Evaluation%20of%20the%20Performance%20of%20Seven%20Antigen%20Rapid%20Diagnostic%20Tests.xlsx.

**Funding:** The studies were largely funded by the providers of these test kits. We acknowledge the Uganda Government funding to UVRI. The procurement of the SARS-CoV2 qRT-PCR kits was funded mainly through the WHO and Africa CDC by different donors, Jack Ma Foundation and the UK Medical Research Council (MRC)/UK Research and Innovation. Part of the funding from the UK Medical Research Council (MRC) and the UK Department for International Development (DFID) is under the MRC/DFID Concordat Agreement and is also part of the EDCTP2 programme supported by the European Union. The funders had no role in study design, data collection and analysis, decision to publish, or preparation of the manuscript.

**Competing interests:** RD is a consultant at Abbott. Therefore he did not participate in the investigation of Panbio™ COVID-19 Ag Rapid test performance. This does not alter our adherence to PLOS ONE policies on sharing data and materials.

## Conclusions

This evaluation identified one Ag RDT, Panbio™ COVID-19 Ag with a performance at high viral load (Ct value $\leq 29$) reaching that recommended by WHO. This kit was recommended for screening of patients with COVID -19-like symptoms presenting at health facilities.

## Introduction

Coronavirus disease 2019 (COVID-19) was first confirmed in Uganda in March 2020 using real-time PCR reverse transcription polymerase chain reaction (qRT-PCR) tests. Over time, the number of specimens tested and the number of COVID-19 cases have increased with a second wave of the epidemic starting in May 2021, and by 6th October 2021, the estimated cumulative PCR tests performed was 1,680,863 while the cumulative number of COVID-19 cases in Uganda stood at 123,445 with 3152 cumulative deaths giving a Case Fatality Rate of 1.1% [1]. However, it is unlikely that all cases were detected since not all suspected cases were tested [1]. There is an increasing demand for testing and sometimes this leads to increased turn-around times (72 hours instead of the desired 24 hours). The qRT-PCR tests are expensive (50–65 USD per test) when personal protective equipment and labour costs are included. In a search for cheaper diagnostic tests and tests that decrease the turn-around time for reporting results compared to qRT-PCR tests [2], the Ministry of Health requested the Uganda Virus Research Institute (UVRI), a COVID-19 national, Africa CDC and WHO reference laboratory to evaluate the performance of COVID-19 Ag RDTs. UVRI had previously evaluated and reported on one Ag RDT, the STANDARD Q COVID-19 Ag Test [3]. Here we provide a performance report for an additional seven Ag RDTs.

The seven RDTs that were evaluated were; BIOCREDIT COVID -19 Ag (RapiGEN, INC, Gyeonggi-do, Korea), COVID-19 Ag Respi-Strip (Coris BioConcept, Gembloux, Belgium), PCL COVID 19 Ag Rapid FIA (Inc, Geumcheon-gu, Seoul, Korea)), MEDsan® SARS-CoV-2 Antigen Rapid test (MEDsan®, Hamburg, Germany), Panbio™ COVID-19 Ag Rapid Test (Abbott Rapid Diagnostics Jena, Germany), Novegent COVID-19 Antigen Rapid Test Kit (colloidal gold) (Chongqing Novegent Biotech Co., Ltd, Chongqing, China) and VivaDiag™ SARS-CoV-2 Ag Rapid Test kit (VivaCheck Biotech(Hangzhou) Co., Ltd. Hangzhou, China). According to the manufacturers' 'Information for Use' (IFU) of the different kits, these Ag RDTs were designed to directly detect SARS-CoV-2 antigens in respiratory secretions. In addition, with the exception of PCL, the other six Ag RDTs evaluated were Conformité Européene (CE) marked (S1 Table).

Although manufacturer's reported high sensitivity (60.0–100%) and specificity (97.8–100%) for their Ag RDTs (S1 Table), the limited peer reviewed reports on their performance in clinical settings [4–8] showed conflicting performance.

The objective of this study was to evaluate the performance of the above Ag RDTs in clinical settings as compared to qRT PCR for detecting SARS-CoV-2 virus in nasopharyngeal samples in order to recommend Ag RDTs that can be used for COVID-19 diagnosis in Uganda.

## Materials and methods

### Study sites, study design, and implementation

This was a prospective cross-sectional study from August 2020 to March 2021. The study was carried out at national and regional COVID-19 isolation centres, established by Ministry of

Health, that received patients from all over the country and at points of entry into the country. The study enrolled travellers requiring testing at points-of-entry (POE) into the country and patients admitted at isolation centres.

At isolation centers we used results from previous PCR tests and records of symptoms in the files to identify participants for recruitment. While at points of entry, participants with and without COVID-19 symptoms were enrolled. Patients that had difficulty in breathing and were receiving oxygen therapy or were on ventilators were excluded from the evaluation.

Enrolment of participants was convenient and consecutive from the selected sites.

Data were collected from each of the participants and entered in an Excel sheet. Data collected included socio-demographic (age, sex, health unit of isolation or point-of-entry), presence or absence of symptoms, date of admission and date of first symptom(s).

### Sample collection and testing

The evaluation of each Ag RDT was performed at different time periods using different sets of samples drawn from participants enrolled at that time. This was because RDTs were received at UVRI from suppliers at different times and in different quantities.

Two nasopharyngeal samples were collected: one from each nostril. One sample was tested on the Ag RDT under evaluation and the second sample was used for qRT- PCR test. The sequence of collection of the nasopharyngeal samples for the Ag RDT and qRT- PCR test was alternated. Samples for qRT-PCR and COVID-19 Ag Respi-Strip were collected in a tube containing virus transport media (Hanks balanced salt solution, Fetal bovine serum, gentamycin sulfate and amphotericin B). The samples for the other six Ag RDT (BIOCREDIT COVID -19 Ag, PCL COVID 19 Ag Rapid FIA, MEDsan® SARS-CoV-2 Antigen Rapid test, Panbio™ COVID-19 Ag Rapid Test, Novegent COVID-19 Antigen Rapid Test Kit (colloidal gold), and VivaDiag™ SARS-CoV-2 Ag Rapid Test kit) under evaluation were placed in the buffer provided by the kit manufacturer.

Antigen RDT testing was done within one hour of specimen collection without any cooling, following the manufacturer's instructions. The results of the six Ag RDTs: BIOCREDIT COVID -19 Ag, COVID-19 Ag Respi-Strip, MEDsan® SARS-CoV-2 Antigen Rapid test, Panbio™ COVID-19 Ag Rapid Test, Novegent COVID-19 Antigen Rapid Test Kit (colloidal gold), and VivaDiag™ SARS-CoV-2 Ag Rapid Test kit were visually read by trained UVRI laboratory staff. While, results from the PCL COVID19 Ag Rapid FIA were not visible and thus only read using the PCLOK EZ instrument supplied with the test kit. Interpretation of Antigen RDT results were as per the respective manufacturer's guidelines. Those reading/interpreting the RDT results sometimes had access to the clinical information of the participant.

Samples collected for qRT-PCR were stored at 2˚-8˚C and transported within one week of collection to UVRI. At UVRI, samples were stored at -80˚C for 1 to 28 days (median of 4–7 days) for the six RDTs except for RESPI Strip which ranged 1–37 days (median of 21 days) before qRT-PCR testing. The personnel carrying out the qRT-PCR testing were blinded to the Antigen RDT results. qRT-PCR testing was carried out at the EPI laboratory, one of the UVRI laboratories that tests for SARS-CoV-2, as described below:

**RNA extraction.** Viral RNA was extracted from the samples using the QIAGEN QIAamp Viral RNA extraction kit (QIAGEN, Hilden, Germany) following the manufacturer's guidelines.

**Real-time reverse-transcription PCR.** qRT-PCR testing was carried out using the Charite-Berlin Protocol [9]. This assay was selected as the Standard reference for the kit evaluation because the protocol has good sensitivity with a limit of detection (LoD) of 3.9 RNA copies per reaction for E-gene assay and 3.6 RNA copies per reaction for RdRp gene assay using

invitro transcribed RNA identical to 2019 novel coronavirus sequences and specificity of 100% [9].

Screening for SARS-CoV viruses was done using the SuperScript™ III Platinum™ One-Step qRT-PCR kit (Invitrogen, Carlsbad, CA) and LightMix® SarbecoV E-gene primer/probe mix (TIB MOLBIOL, Berlin, Germany) following the manufacturer's guidelines. All samples that had Ct values <45 in the above assay were subjected to a SARS-CoV-2 confirmatory qRT-PCR using the SuperScript III real-time RT-PCR kit (Invitrogen, Carlsbad, CA) and LightMix® Modular SARS-CoV-2 (COVID-19) RdRp primer/probe mix (TIB MOLBIOL, Berlin, Germany) following the manufacturer's guidelines. A sample was considered positive if it had a Ct value ≤39 on the confirmatory qRT-PCR. A sample was considered negative if the Ct value was undetermined on the screening qRT-PCR or if it was positive on the screening qRT-PCR but undetermined or >39 on the confirmatory qRT-PCR.

The thermal cycling conditions consisted of a reverse transcription reaction at 50°C for 10 minutes, an activation step at 95°C for 10 minutes and 45 cycles at 95°C/15 seconds and 60°C /1 minute. The platform used for the PCR was Applied Biosystems:7500 Real-Time PCR System (Marsiling, Singapore).

## Data management and analysis

Ct values were recorded against each sample's identifier and data entered in an MS Excel sheet which was transformed into a STATA® v15 (StataCorp LP, 4905 Lakeway Drive, College Station, TX, USA) file for analysis. Inconsistency checks were performed using STATA and data cleaning done in collaboration with the field and laboratory staff. Samples with missing Antigen RDT and qRT-PCR results were excluded from the analysis.

Univariate analysis was performed to generate descriptive summaries for demographic and sample source characteristics. Frequencies, means, confidence intervals and medians were generated as summary statistics.

## Performance evaluation

**Sensitivity.** Sensitivity was calculated as the number of specimens determined as positive by the Ag RDT under evaluation divided by the number of specimens determined as positive by PCR and expressed as a percentage with confidence intervals.

**Specificity.** Specificity was calculated as the number of specimens determined as negative by the Ag RDT under evaluation divided by the number of specimens determined as negative by PCR and expressed as a percentage with confidence intervals.

**Accuracy.** The accuracy was calculated as the proportion of results determined by the Ag RDT under evaluation that agreed with the PCR results and expressed as a percentage with confidence intervals.

**False positive rate.** The false positive rate was calculated as False Positive/(False Positive + True Negative) and expressed as a percentage.

**False negative rate.** The false negative rate was calculated as False Negative/(False Negative + True Positive) and was expressed as a percentage.

Sensitivity, specificity, accuracy, false positive rate and false negative rate calculations were performed using the proportion command in STATA 15 and confidence intervals produced with the Wilson score method [10].

Sensitivity of the Ag RDTs was also determined for different Ct cut-off values of ≤29, ≤33 and ≤39.

Sensitivity and specificity for all the Ag RDTs were also determined according to whether symptom onset was within 7 days of the specimen collection or after 7 days.

A sub-analysis (for sensitivity) was also performed for each Ag RDT considering different Ct cut off values following discussions with relevant stakeholders at the Ministry of Health and basing on literature [11–13]; a strong positive was defined as Ct value ≤29, a moderate positive was defined as Ct value 30–37 and a low positive was defined as Ct value 38–39.

### Ethical considerations

The study protocol [14] was approved by Uganda Virus Research Institute's Research Ethics Committee (No. GC/127/20) and the Uganda National Council for Science and Technology (No HS637ES). Informed written consent was obtained from each of the participants before enrolment into the study.

## Results

### Participant demographics

A total of 1,533 participants were included in the evaluation of the seven Ag kits as detailed in Table 1 below. The mean age group was 36 years, and 1024(66.8%) were males, 493 (32.2%) were females while 16(1.0%) had missing information on gender.

### BIOCREDIT COVID -19 Ag test

**Participant characteristics.** Samples for the evaluation of the BIOCREDIT COVID-19 Ag RDT were collected in August 2020, from 247 participants (S1 Fig) with 122 (49.4%) from

**Table 1. Demographic characteristics of participants.**

| Antigen RDT | BIOCREDIT COVID -19 Ag | COVID-19 Ag Respi-Strip | PCL COVID19 Ag Rapid FIA | MEDsan® SARS-CoV-2 Antigen Rapid test | Panbio™ COVID-19 Ag Rapid test | Novegent COVID-19 Antigen Rapid test kit (colloidal gold) | VivaDiag™ SARS-CoV-2 Ag Rapid Test kit |
|---|---|---|---|---|---|---|---|
| N | 247 | 194 | 172 | 243 | 185 | 229 | 263 |
| **Sex** | | | | | | | |
| M | 165 (66.8) | 156 (80.4) | 142(82.6) | 156 (64.2) | 136(73.5) | 132(57.6) | 137 (52.1) |
| F | 76 (30.8) | 38 (19.6) | 30 (17.4) | 77 (31.7) | 49(26.5) | 97(42.4) | 126 (47.9) |
| Missing | 6 (2.4) | | | 10 (4.1) | | | |
| **Age in years** | | | | | | | |
| <20 | 2(0.8) | 4(2.1) | 3(1.7) | 5(2.1) | 6(3.2) | 1(0.4) | 4(1.5) |
| 20–29 | 68(27.5) | 54(27.8) | 49(28.5) | 39(16.0) | 53(28.7) | 28(12.2) | 77(29.3) |
| 30–39 | 87(35.2) | 62(32.0) | 48(27.9) | 39(16.0) | 59(31.9) | 52(22.7) | 66(25.1) |
| 40–49 | 29(11.8) | 34(17.5) | 22(12.8) | 36(14.8) | 33(17.8) | 46(20.1) | 41(15.6) |
| 50 and above | 61(24.7) | 17(8.8) | 50(29.1) | 41(16.9) | 34(18.4) | 97(42.4) | 71(27.0) |
| Missing | | 23(11.9) | | 83(34.2) | | 5 (2.2) | 4 (1.5) |
| **Duration of symptoms** | | | | | | | |
| 0–7 Days | 97(39.3) | No information | 32(18.6) | 68(28.0) | 117(63.2) | 137(59.8) | 235(89.4) |
| >7 Days | 42(17.0) | | 11(6.4) | 49(20.2) | 20(10.8) | 88(38.4) | 21(8.0) |
| No symptoms/ missing | 108(43.7) | | 129(75.0) | 126(51.9) | 48(26.0) | 4(1.8) | 7(2.7) |
| **PCR (Berlin protocol)** | | | | | | | |
| Positive | 135 (54.7) | 67 (34.5) | 93 (54.1) | 123 (50.6) | 85 (46.0) | 100(43.7) | 43 (16.4) |
| Negative | 112 (45.3) | 127 (65.5) | 79 (45.9) | 120 (49.4) | 100(54.1) | 129(56.3) | 220 (83.6) |

**Table 2. Field performance of 7 antigen RDTs for the diagnosis of SARs-CoV-2 virus in Uganda.**

| Kit name | Total participants recruited | Test sensitivity (%) Ct $\leq$ 39 (95% CI) | Test sensitivity (%) Ct $\leq$ 33 (95% CI) | Test sensitivity (%) Ct $\leq$ 29 (95% CI) | Test specificity % (95% CI) | RDT accuracy (%) Ct $\leq$ 39 (95% CI) |
|---|---|---|---|---|---|---|
| BIOCREDIT COVID -19 Ag | 247 | 27.4 (20.5%–35.6%) | 44.7 (33.8%–56.3%) | 60 (45.5%–72.9%) | 98.2 (93.1%–99.6%) | 59.8 (53.6%–65.8%) |
| COVID-19 Ag Respi-Strip | 194 | 19.4 (11.5%–30.9%) | 35.1 (21.1%–52.4%) | 46.4 (28.2%–65.7%) | 99.2 (94.5%–99.9%) | 71.6 (64.8%–77.6%) |
| PCL COVID19 Ag Rapid FIA | 172 | 37.6 (28.2%–48.1%) | 71.0 (51.8%–84.8%) | 77.8 (57.0%–90.2%) | 89.9 (80.8%–94.9%) | 61.6 (54.1%–68.7%). |
| MEDsan® SARS-CoV-2 Antigen Rapid test | 243 | 13.0 (8.1%–20.3%) | 23.7 (44.7%–74.3%) | 43.5 (24.0%–65.3%) | 100 (96.9%–100%) | 56.0 (49.6%–62.1%) |
| Panbio™ COVID-19 Ag Rapid test | 185 | 49.4 (38.7%–60.1%) | 72 (57.6%–83.0%) | 85.7 (66.0%–94.9%) | 100 (96.4%–100%) | 76.8 (70.1%–82.3%) |
| Novegent COVID-19 Antigen Rapid test kit (colloidal gold) | 229 | 46 (36.3%–56.0%) | 58.2 (44.5%–70.8%) | 66.7 (46.0%–82.5%) | 89.9 (83.3%–94.1%) | 70.7 (64.5%–76.3%) |
| VivaDiag™ COVID-19 Ag | 263 | 30.2 (18.0%–46.1%) | 31.6 (18.4%–48.6%) | 80 (37.8%–96.3%) | 94.1 (90.1%–96.6%) | 83.7 (78.6%–87.7%) |

Mulago National Referral Hospital (NRH), 27 (10.9%) from Entebbe Regional Referral Hospital (RRH) and 98 (39.7%) from Namboole National Stadium isolation centre.

One hundred thirty-five participants (54.7%) were PCR-positive with 48 (36.1%) of these being females. The median age of the participants was 33 years (IQR 28–39 years).

There were 97 participants with onset of symptoms ranging between 0–7 days prior to the evaluation, 42 with onset of symptoms more than 7 days prior to the evaluation and 108 participants who were asymptomatic before the Ag RDT test (Table 1).

**BIOCREDIT COVID -19 Ag test performance.** There were 39 (15.7%) specimens that were BIOCREDIT COVID -19 Ag test positive (S1 Fig). The sensitivity of the test was 27.4% (95% CI: 20.5%– 35.6%) at $\leq$39 Ct cut-off, 44.7% (95% CI: 33.8–56.3%) at Ct cut-off values $\leq$33 and 60.0% (95% CI: 45.5%-72.9%) at Ct cut-off values $\leq$29 (Table 2).

Among the participants with symptom onset to date of Ag RDT test between 0–7 days, the sensitivity of the test was 39.3% (95% CI: 27.7%– 52.4%). The sensitivity of the test was 11.1% (3.3%–31.1%) for participants with more than 7 days from symptom onset to date of Ag RDT test: and 21.3% (95% CI: 11.6%–35.8%) for asymptomatic participants.

The Ag RDT specificity was 98.2% (95% CI: 93.1%–99.6%). The accuracy of the test was 59.8% (95% CI: 53.6%–65.8%) (Table 2); The false positive rate (FPR) was 5.1% (95% CI: 1.2%– 19.3%) and the false negative rate (FNR) was 72.6% (95% CI: 64.4%– 79.5%) (Table 2).

The BIOCREDIT COVID -19 Ag test result was negative for a large proportion of strong, moderate, and low positive specimens by qRT-PCR: 40%, 90% and 100% respectively. This was observed especially for Ct cut-off values between 30–38 (Table 3).

## COVID 19 antigen respi-strip test (Respi-Strip)

**Participant characteristics.** Samples for the Respi-Strip evaluation were collected in August 2020, from 194 participants (S2 Fig) with 84 (43.3%) from Mulago NRH, 71 (36.6%) from Malaba POE, 33 (17.0%) from Entebbe RRH and 6 (3.1%) from Mbale RRH. Seventy- six (66) participants had COVID-19 Ag Respi-Strip and PCL COVID19 Ag Rapid FIA. While fifteen (15) participants had both COVID-19 Ag Respi-Strip and BIOCREDIT COVID -19 Ag Test and One hundred and three (103) had only COVID-19 Ag Respi-Strip tests.

**Table 3. COVID-19 Antigen Rapid test results compared to the Ct values on RT-PCR.**

| COVID-19 Antigen kit results | | qRT-PCR Ct Value | | |
|---|---|---|---|---|
| Kit | Results | Strong Positive (≤29) N (%) | Moderate Positive (30–37) N (%) | Low Positive (38–39) N (%) |
| BIOCREDIT COVID-19 Ag | Positive | 30 (60) | 7 (9.6) | 0 (0.0) |
| | Negative | 20 (40) | 66 (90.4) | 12 (100) |
| PCL COVID-19 | Positive | 21 (77.8) | 12 (20.7) | 2 (25.0) |
| | Negative | 6 (22.2) | 46 (79.3) | 6 (75.0) |
| Respi-Strip COVID-19 Ag | Positive | 13 (46.4) | 0 (0) | 0 (0) |
| | Negative | 15 (53.6) | 37 (100) | 2 (100) |
| MEDsan® SARS-CoV-2 Antigen Rapid test | Positive | 10 (43.5) | 6 (6.5) | 0 (0.0) |
| | Negative | 13 (56.5) | 87 (93.6) | 7 (100) |
| Panbio™ COVID-19 Ag Rapid test | Positive | 24 (85.7) | 17 (30.9) | 1 (50.0) |
| | Negative | 4 (14.3) | 38 (69.1) | 1 (50.0) |
| Novegent COVID-19 Antigen Rapid test kit (colloidal gold) | Positive | 18 (66.8) | 27 (40.3) | 1 (16.7) |
| | Negative | 9 (33.3) | 40 (59.7) | 5 (83.3) |
| VivaDiag™ COVID-19 Ag | Positive | 8 (80.0) | 5 (15.6) | |
| | Negative | 2 (20.0) | 27 (84.4) | |

Sixty-seven (34.5%) participants out of the 194 were positive on PCR with 6 (9.0%) of these positives being females. The median age was 33 years (IQR 28–41 years).

**Respi-Strip test performance.** There were 14 (7.2%) specimens that were Respi-Strip test positive (S2 Fig). The test showed a sensitivity of 19.4% (95% CI: 11.5%– 30.9%) at Ct values ≤39; 35.1% (95% CI: 21.1–52.4%) at Ct values ≤33 and 46.4% (95% CI: 28.2%-65.7%) at Ct values ≤29 (Table 2). The specificity of the test was 99.2% (95% CI: 94.5%–99.9%).

The accuracy of the Respi-Strip test was 71.6% (95% CI: 64.8%–77.6%) at Ct values ≤39 (Table 2); the FPR was 0.8% (95% CI: 0.1%– 5.5%); and the FNR was 80.6% (95% CI: 69.1%–88.5%) (Table 2).

The Respi-Strip determined the largest proportion of strong positives by qRT-PCR as negative (53.6%) and all moderate and low positives by qRT-PCR as negative (100%) (Table 3).

## PCL COVID19 Ag Rapid FIA (PCL)

**Participant characteristics.** Samples for the evaluation were collected between August and September 2020, from 172 participants (S3 Fig) with 155 (90.1%) from Mulago NRH and 17 (9.9%) from Entebbe RRH. Ninety-three participants (54.1%) were PCR positive with 12 (12.9%) of these being females. The median age was 32 years (IQR 28–39 years).

There were 32 participants with onset of symptoms to date of Ag RDT test between 0–7 days. Of these, 19 were PCR-positive. Another 15 participants reported symptom onset of more than 7 days prior to the Ag RDT test with 9 (60%) being qRT-PCR positive (Table 1).

**PCL Ag Rapid FIA test performance.** There were 43 (25%) specimens that were Ag RDT-positive (S3 Fig). Overall, the PCL Ag test had a sensitivity of 37.6% (95% CI: 28.2%–48.1%) at Ct values ≤39: 71.0% (95% CI: 51.8%– 84.8%) at Ct values ≤33 and 77.8% (95% CI: 57.0%– 90.2) at Ct values ≤29 (Table 2).

The sensitivity of the Ag RDT was 26.3% (95% CI: 10.4%– 52.4%) for participants with symptom onset to date of test within 0–7 days and 11.1% (0.9%–62.6%) for those with symptom onset to date of test greater than 7 days. For asymptomatic participants, sensitivity of the Ag RDT was 46.0% (95% CI: 33.9%–58.7%).

The specificity of the test was 89.9% (95% CI: 80.8%–94.9%) (Table 2). The accuracy of the Ag RDT was 61.6% (95% CI: 54.1%–68.7%) (Table 2). The FPR was 10.1% (95% CI: 5.1%–19.2%) and the FNR was 62.4% (95% CI: 51.9%– 71.8%) (Ct cut-off ≤39) (Table 2).

The PCL test determined a large proportion of samples, whether strong or moderate or low positive by qRT-PCR as negative: 22.2%, 79.3% and 75.0% respectively (Table 3).

## MEDsan® SARS-CoV-2 Antigen Rapid test

**Participant characteristics.** A total of 243 samples (S4 Fig) were collected in November and December 2020 (from 228 participants; some participants gave multiple samples) with 137 samples (56.4%) from Mulago NRH, 10 (4.1%) from Entebbe RRH and 96 (39.5%) from Namboole National Stadium isolation center.

Of the 243 samples, 123 (50.6%) were qRT-PCR positive: 37 (30.1%) of these positives were females. The median age of the participants was 37 years (IQR 28–49 years). A total of 75(30%) participants had onset of symptoms 0–7 days before the antigen test (Table 1).

There were 25 qRT-PCR positive participants with the period between symptom onset to the date when the antigen test was done ranging between 0–7 days. For twenty- three (23) participants the period was greater than 7 days.

**MEDsan® SARS-CoV-2 Antigen Rapid test performance.** There were 16 (6.6%) specimens that were MEDsan® SARS-CoV-2 Antigen Rapid test positive (S4 Fig). The MEDsan® SARS-CoV-2 Antigen Rapid test showed a sensitivity of 13.0% (95% CI: 8.1%– 20.3%) at a Ct cut off ≤39; a sensitivity of 23.7% (44.7–74.3%) at Ct values ≤33 and sensitivity of 43.5% (24.0–65.3%) at Ct values ≤29 (Table 2). The Ag RDT had a sensitivity of 12.0% (95% CI: 3.96%– 33.3%) for participants with 0–7 days between symptom onset and the antigen test and 21.7% (6.7%–44.8%) for participants with more than 7 days between symptom onset and the antigen test.

The specificity of the test was 100% (95% CI: 96.9%–100%) (Table 2).

The accuracy of MEDsan® SARS-CoV-2 Antigen Rapid test was 56.0% (95% CI: 49.6%–62.1%) (Table 2). The FPR was 0.0% (95% CI: 0.0%– 19.4%) and the FNR was 87.0% (95% CI: 79.7%– 91.9%).

The MEDsan® SARS-CoV-2 Antigen Rapid test determined a big proportion of strong, moderate and weak positives by qRT-PCR as negative, i.e. 56.5%, 93.6% and 100% respectively (Table 3).

## Panbio™ COVID-19 Ag Rapid test

**Participant characteristics.** One hundred eight five (185) participants (S5 Fig) were recruited in October 2020, 133 (71.9%) from Namboole Stadium isolation center, 40 (21.6%) from Mulago NRH and 12 (6.5%) from Entebbe RRH. Out of the 185 participants, 85 were qRT-PCR positive and 100 were qRT-PCR negative. One hundred seventeen (117(63.2%)) participants had a range of 0–7 days from symptom onset to the date when the antigen test was done (Table 1). Of these 117, forty-five (45) were positive by qRT-PCR. Twenty (20) participants had symptom onset more than 7 days prior to the test, and 9 out of the 20 (45%) tested positive by qRT-PCR.

The majority of the participants were males (73.5%) and the median age was 34 years (IQR 28–44 years) (Table 1).

**Panbio™ COVID-19 Ag Rapid test performance.** There were 42 (22.7%) specimens that were Panbio™ COVID-19 Ag Rapid test positive (S5 Fig). The Panbio™ COVID-19 Ag Rapid test showed a sensitivity of 49.4% (95% CI: 38.7%– 60.1%) (Table 2) when compared to qRT-PCR test results at a Ct cut off value ≤ 39; a sensitivity of 72% (95% CI: 57.6%–83.0%) at

Ct values ≤33 and 85.7% (95% CI: 66.0%–94.9%) at Ct values ≤29 (Table 2). The sensitivity of the Panbio™ COVID-19 Ag Rapid test was 51.1% (95% CI: 36.3%– 65.8%) for participants with symptom onset 0–7 days before the antigen test while the sensitivity of the antigen test was 44.4% (13.4%–80.5%) for participants with symptom onset greater than 7 days before the antigen test. In asymptomatic participants, the sensitivity of the test was 48.4% (95% CI: 30.8%–66.4%).

The test had a specificity of 100% (95% CI: 96.4%–100%) (Table 2).

The Panbio™ COVID-19 Ag Rapid test showed an accuracy of 76.8% (95% CI: 70.1%–82.3%) (Table 2). The FPR was 0% (95% CI: 0%–8.4%) and the FNR was 50.6% (95% CI: 39.9%– 61.2%).

It was observed that the Panbio™ COVID-19 Ag Rapid test is more likely to determine a sample as positive when a sample has abundant target nucleic acid with Ct ≤ 29 (85.7%). However, when a sample is determined as moderate positive, 69% were determined as negative by Panbio™ COVID-19 Ag Rapid test (Table 3).

## Novegent COVID-19 antigen rapid test kit (colloidal gold)

**Participant characteristics.** A total of 229 participants (S6 Fig) were included in the evaluation: 32 (14.0%) participants from Entebbe RRH, 111 (48.5%) from Namboole National Stadium isolation center, 85 (37.1%) from Mulago RRH and 1 (0.4%) from Kisubi Hospital. The samples for evaluation were collected between January and February 2021. Of these 229 partcipants, 100 were qRT-PCR positive and 129 were qRT-PCR negative (Table 1).

There were 137 participants with a period of 0–7 days between symptom onset and the date of the antigen test and 56 of these were qRT-PCR positive. Forty- two (42) qRT-PCR positive participants had symptom onset more than 7 days prior to the test (Table 1).

The majority of the participants were males (57.6%) and the median age was 47 years (IQR 34–58 years) (Table 1).

**Novegent COVID-19 Antigen Rapid test kit (colloidal gold) performance.** There were 59 (25.8%) participants that were Novegent COVID-19 Antigen Rapid test positive (S6 Fig).

The test showed a sensitivity of 46% (95% CI: 36.3%– 56.0%) when compared to qRT-PCR test results at a Ct cut off value of ≤ 39;a sensitivity of 58.2% (95% CI: 44.5%–70.8%) at a Ct cut off value of ≤33 and sensitivity of 66.7% (95% CI: 46.0%–82.5%) at a Ct cut-off value of ≤29 (Table 2).

Taking into consideration participants with 0–7 days between symptom onset and the date of the antigen test, the Antigen RDT sensitivity was 65.0% (95% CI: 38.6%– 71.4%). For those participants where the period between symptom onset and the date of the antigen test was greater than 7 days the RDT sensitivity was 42.9% (28.4%–58.7%).

The test had a specificity of 89.9% (95% CI:83.3%–94.1%) (Table 2).

The RDT accuracy was 70.7% (95% CI: 64.5%–76.3%) (Table 2): the FPR was 22.0% (95% CI:13.0%–34.8%) and the FNR was 31.8% (95% CI: 25.1%– 39.2%) (Table 2).

Novegent COVID-19 Antigen Rapid test (Colloidal) is more likely to determine a sample as positive (66.7%) if the qRT-PCR result is a strong positive. However, for those samples that were moderately or weakly qRT-PCR positive, this antigen test was more likely to determine them as negative, 59.7% and 83.3% respectively (Table 3).

## VivaDiag™ SARS-CoV-2 Antigen Rapid test

**Participant characteristics.** In January to March 2021, samples for the evaluation were collected from 263 participants (S7 Fig): 122 (46.4%) from Kiruddu NRH, 59 (22.4%) from CASE Hospital, 77 (29.3%) from Mulago NRH and 5 (1.9%) from Entebbe RRH. Forty- three

(16.4%) were positive on qRT-PCR with 21 (48.8%) of these positives being females. The median age of the participants was 36 years, (IQR 28–51 years) (Table 1).

There were 235(89.3%) participants with a period ranging between 0–7 days from symptom onset to the date when the antigen test was done(Table 1). Of these 235, thirty-eight (38) were positive by qRT-PCR. Twenty-one participants reported symptom onset more than 7 days prior to the antigen test with 5 (23.8%) determined as qRT-PCR-positive.

**VivaDiag™ SARS-CoV-2 Antigen Rapid test performance.** There were 26 (7.8%) participants that were VivaDiag™ antigen test positive (S7 Fig).

The VivaDiag™ Ag test showed a sensitivity of 30.2% (95% CI: 18.0%– 46.1%) at qRT-PCR Ct values ≤39; a sensitivity of 31.6% (95% CI: 18.4%–48.6%) at qRT-PCR Ct values ≤33 and a sensitivity of 80% (95% CI: 37.8%–96.3%) at qRT-PCR Ct values ≤29 (Table 2). Taking into consideration participants with symptom onset 0–7 days prior to the antigen test, the sensitivity of the antigen test was 26.3% (95% CI: 14.4%– 43.2%).

The specificity of the test was 94.1% (95% CI: 90.1%–96.6%) (Table 2).

The VivaDiag™ Ag test showed an accuracy of 83.7% (95% CI: 78.6%–87.7%) (Table 2); The FPR was 50.0% (95% CI: 30.5%– 69.5%); and the FNR was 12.7% (95% CI: 9.0%– 17.6%).

The VivaDiag™ determined a large proportion of qRT-PCR strong positive cases as positive (80%) though the number of cases in this category was only five. However, VivaDiag™ determined a very large proportion of the qRT-PCR moderately-positive cases (Ct values 30–38) as negative (84.4%) (Table 3).

## Discussion

WHO recommends the use of antigen RDTs for SARS-CoV-2 diagnosis because they provide results within 15–30 minutes compared to qRT-PCR tests where results may not be available for more than 24 hours. RDTs cost far less than PCR tests, and do not require specialized laboratories and highly trained staff [2]. The use of accurate COVID -19 antigen RDTs will enable faster identification and management of COVID-19 patients leading to better control of the pandemic. A large number of antigen RDTs have been developed with varying sensitivities and specificities reported by the manufacturer's IFUs. However, WHO has recommended field evaluations of antigen RDTs before they can be adopted in a national setting [2]. WHO recommends the use of antigen RDTs with a sensitivity of ≥80% and a specificity of ≥ 97% [2].

In this study, none of the seven antigen RDTs evaluated reached the sensitivity reported by the manufacturers, even after considering samples strongly positive by qRT-PCR (Ct values ≤29). These RDTs however had good specificity, with only three showing a specificity slightly lower than that reported in the IFUs. Poor performance in real-world settings compared to that found in the manufacturer's IFUs has also been reported by others [4,7,8,15–18], emphasizing the need for performance evaluations of antigen RDTs in-country before adoption for use. In our study, we noted that the sensitivity of all seven Ag RDTs improved at lower qRT-PCR Ct values and this has also been observed in other studies [6–8,18–20]. Lower qRT-PCR Ct values in most cases correlate with higher viral loads and hence with increased transmissibility of the virus, usually in the early phases of the infection [21].

There were two antigen RDTs that reached the WHO recommended sensitivity performance of ≥80% at a Ct value of ≤29. These were the Panbio™ COVID-19 Ag Rapid test and the VivaDiag™ SARS-CoV-2 Ag Rapid test with sensitivities of 85.7% and 80% respectively. However, for the VivaDiag, the sample size was very small with only 10 samples at this Ct value. Of the two kits, Panbio™ COVID-19 Ag Rapid test and VivaDiag™ SARS-CoV-2 Ag

Rapid test, only the PanBio™ COVID-19 Ag Rapid Test had the WHO recommended specificity performance of $\geq$ 97%.

There was no correlation of antigen RDT performance with the presence or absence of symptoms. High viral loads usually appear in the pre-symptomatic and early symptomatic phases of the illness (within the first 5–7 days of symptom onset), however asymptomatic individuals can also have high viral loads in the early days of infection [22].

The Panbio™ COVID-19 Ag Rapid Test met the sensitivity and specificity performance levels recommended by WHO at qRT-PCR Ct values of $\leq$29. This antigen RDT together with the Standard Q (by SD Biosensor) antigen RDT we previously evaluated [3], have been recommended in Uganda for COVID-19 diagnostic intervention in a phased approach as more experience and confidence is gained in their use through continuous field evaluation and the generation of additional data. These antigen RDTs have been recommended for rapid screening of the following populations; symptomatic alerts and symptomatic contacts of confirmed cases, patients with COVID -19-like symptoms presenting at health facilities.

The Uganda MOH recommends that, any antigen RDT positive case from the above mentioned populations will be considered "COVID– 19 positive" and managed accordingly and will not require additional qRT-PCR confirmation except in special cases such as genomic sequencing or routine quality control monitoring. Furthermore, any antigen RDT negative case with highly suggestive symptoms will be considered "a suspect COVID– 19 case" until confirmed to be uninfected by qRT-PCR and should be managed with enhanced infection prevention control (IPC). Laboratories at health units are expected to obtain/collect an additional specimen from those symptomatic clients who are antigen RDT negative to be sent for PCR testing.

## Study limitations

Since samples were collected from different nostrils for the antigen test (BIOCREDIT COVID -19 Ag, MEDsan® SARS-CoV-2 Antigen Rapid test, PCL and Panbio™ COVID-19 Ag Rapid test) and qRT-PCR tests, the two swabs can be considered as two different samples which may lead to variation in the virus content of the collected samples.

The Panbio™ COVID-19 Ag Rapid Test IFU recommends using the test within 5 days of symptom onset. However, in the evaluation of this antigen RDT, specimens from some asymptomatic cases were included. In asymptomatic cases it is difficult to determine the stage of the infection, whether early or late, thus affecting the overall interpretation of the results i.e. whether the antigen test result is negative because the client was in the early stage of disease and thus had low levels of antigen or was due to the poor performance of the test kit.

Participants in the study were recruited at different time periods thus there was varying prevalence of SARS-CoV-2 which could have affected the specificity of the tests. There was also significant difference in the population mix of the participants for the different tests (male/female, mean age, number of PCR positive and PCR negative participants) which may affect the comparability of the tests. Most participants were tested with only one of the evaluated RDT, implying different populations were used for evaluation of each RDT, thus making it difficult to compare the different RDT performance.

## Conclusions

This field evaluation of seven SARS CoV-2 RDTs: COVID-19 Ag Respi-Strip, BIOCREDIT COVID -19 Ag, MEDsan® SARS-CoV-2 Antigen Rapid test, PCL COVID19 Ag Rapid FIA, Panbio™ COVID-19 Ag Rapid test, Novegent COVID-19 Antigen Rapid test kit (colloidal gold), and VivaDiag™ COVID-19 Ag showed poorer performance than that reported by the

manufacturers. Only the Panbio™ COVID-19 Ag Rapid Test reached the WHO recommended performance of a sensitivity of $\geq 80\%$ and a specificity $\geq 97\%$ at qRT-PCR Ct values $\leq 29$. This RDT can be recommended for COVID-19 infection diagnosis among symptomatic patients in health facilities in order not to delay patient management while waiting for qRT-PCR results.

## Supporting information

**S1 Fig. Flowchart summarizing test results using the BIOCREDIT COVID-19 Ag RDT.**
(TIF)

**S2 Fig. Flowchart summarizing test results using the COVID-19 Ag Respi-Strip.**
(TIF)

**S3 Fig. Flowchart summarizing test results using the PCL COVID 19 Ag Rapid FIA.**
(TIF)

**S4 Fig. Flowchart summarizing test results using the MEDsan® SARS-CoV-2 Antigen Rapid test.**
(TIF)

**S5 Fig. Flowchart summarizing test results using the Panbio™ COVID-19 Ag Rapid test.**
(TIF)

**S6 Fig. Flowchart summarizing test results using the Novegent COVID-19 Antigen Rapid test kit.**
(TIF)

**S7 Fig. Flowchart summarizing test results using the VivaDiag™ SARS-CoV-2 Ag Rapid test kit.**
(TIF)

**S1 Table. Principal and kit performance of the seven evaluated COVID -19 Rapid Antigen test kits.**
(DOCX)

## Acknowledgments

We thank Thomas Nsibambi, CDC Uganda for the technical input. We thank the participants who provided samples for the evaluation. We thank all the health workers at Mulago NRH, Kiruddu NRH, Mbale and Entebbe RRHs, CASE and Kisubi Hospitals, Namboole National Stadium isolation center and Malaba POE that collected the samples from the participants.

EPI Laboratory team: [1][**]Prossy Namuwulya, [1]Francis Aine, [1]Irene Turyahabwe, [1]Lucy Nakabazzi, [1]Molly Birungi, [1]Rajab Dhatemwa, [1]Arnold Mugagga, [1]Mayi Tibanagwa, [1]Mathias Ssenono, [1]Charles Okia, [1]Mary Nyachwo.

Author affiliations: [1]: Uganda Virus Research Institute, Plot 51–59 Nakiwogo Road, P.O. BOX 49, Entebbe, Uganda.

[**]Lead author and contacts: Prossy Namuwulya

pnamuwulya@uvri.go.ug

UVRI COVID-19 Technical team: [***] [1]John Kayiwa, [1,2]Jennifer Serwanga, [1]Stephen Balinandi, [1]Christine Watera, [1]Aminah Nalumansi, [1]Jocelyn Kiconco and [1]Bernard Kikaire.

Author Affiliations:

[1] Uganda Virus Research Institute, Plot 51–59 Nakiwogo Road, P.O.Box 49, Entebbe

[2] Medical Research Council/Uganda Virus Research Institute & London School of Hygiene and Tropical Medicine, Uganda Research Unit, P.O. Box 49, Plot 51–59, Nakiwogo Road, Entebbe, Uganda

***Lead author and contacts: John Kayiwa

jkayiwa@yahoo.com

## Author Contributions

**Conceptualization:** Josephine Bwogi, Tom Lutalo, Henry Bukenya, Matthew Cotten, Robert Downing, Julius Lutwama, Pontiano Kaleebu.

**Data curation:** Tom Lutalo.

**Formal analysis:** Tom Lutalo.

**Investigation:** Josephine Bwogi, Phionah Tushabe, Henry Bukenya, James Peter Eliku, Isaac Ssewanyana, Susan Nabadda, Christopher Nsereko.

**Methodology:** Josephine Bwogi, Tom Lutalo, Henry Bukenya, Pontiano Kaleebu.

**Project administration:** Josephine Bwogi.

**Resources:** Matthew Cotten, Pontiano Kaleebu.

**Supervision:** Josephine Bwogi, Phionah Tushabe, Henry Bukenya, Christopher Nsereko, Julius Lutwama, Pontiano Kaleebu.

**Validation:** Isaac Ssewanyana, Susan Nabadda.

**Visualization:** Josephine Bwogi, Tom Lutalo.

**Writing – original draft:** Josephine Bwogi, Tom Lutalo, Phionah Tushabe, Pontiano Kaleebu.

**Writing – review & editing:** Josephine Bwogi, Tom Lutalo, Phionah Tushabe, Henry Bukenya, James Peter Eliku, Isaac Ssewanyana, Susan Nabadda, Christopher Nsereko, Matthew Cotten, Robert Downing, Julius Lutwama, Pontiano Kaleebu.

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
