## [Decision Letter · Decision Letter 0]

10 Sep 2021

PONE-D-21-24051Field Evaluation of the Performance of Seven Antigen Rapid Diagnostic Tests for the Diagnosis of SARs-CoV-2 Virus Infection in UgandaPLOS ONE

Dear Dr. Bwogi,

Thank you for submitting your manuscript to PLOS ONE. After careful consideration, we feel that it has merit but does not fully meet PLOS ONE’s publication criteria as it currently stands. Therefore, we invite you to submit a revised version of the manuscript that addresses the points raised during the review process.

Please attend to the all the concerns that have been raised by all the reviewers. Among some of the things that have been highlighted include:

1. Information in the tables must be well presented and numbers must tally with those that are given in the text. The order of presentation of results in the text must follow the order in which they appear in the tables..

2. Provide more information about the participants in this study and how they were selected. Do they represent the population on which such a the tests are intended to be used on? Was each test applied to every participant and if not, what impact would this have on the comparison of the test results.

3. Provide more information about the validity of the RT-PCR that was used as a Gold Standard in this study. 

We look forward to receiving your revised manuscript.

Kind regards,

Martin Chtolongo Simuunza, PhD

Academic Editor

PLOS ONE

Journal Requirements:

[RD is a consultant at Abbott. Therefore he did not participate in the investigation of Panbio™ COVID-19 Ag Rapid test performance.]. 

5. One of the noted authors is a group or consortium [EPI Laboratory team and UVRI COVID 19 Technical team.] In addition to naming the author group, please list the individual authors and affiliations within this group in the acknowledgments section of your manuscript. Please also indicate clearly a lead author for this group along with a contact email address.

Reviewers' comments:

Reviewer's Responses to Questions

**Comments to the Author**

1. Is the manuscript technically sound, and do the data support the conclusions?

Reviewer #1: Partly

Reviewer #2: Yes

Reviewer #3: Partly

2. Has the statistical analysis been performed appropriately and rigorously? 

Reviewer #1: Yes

Reviewer #2: No

Reviewer #3: Yes

3. Have the authors made all data underlying the findings in their manuscript fully available?

Reviewer #1: No

Reviewer #2: Yes

Reviewer #3: No

4. Is the manuscript presented in an intelligible fashion and written in standard English?

Reviewer #1: Yes

Reviewer #2: Yes

Reviewer #3: Yes

5. Review Comments to the Author

Reviewer #1: The manuscript is interesting and well written.

The authors should acknowledge more limitations of the study. The comparison between the tests is not fair, because the tests are not compared in a randomized way. Sensitivity and specificity can be different if samples come from populations with different disease spectrum/prevalence.

The STARD checklist should be used (https://www.equator-network.org/reporting-guidelines/stard).

Table 1 should completed with percentages for all the variables (e.g. age groups).

Tables should be formatted in a better way.

Is the time lag between Antigen Test and PCR a possible explanation for the difference observed with vendors diagnostic performance?

Reviewer #2: Reviewer Recommendation and Comments for Manuscript Number PONE-D-21-24051

The comparison of the panel of tests is key to improving the use of these tests in local situation. Such is very important and carrying out such investigations improve the performance as mitigation measures can be put in place in real time. The fact that such tests can be performed at the bedside therefore hasten the recovery process and facilitates a speedy intervention and therefore recovery.

Observation

1. There is need for the authors to revisit the figures (number/values) in the text as well as those in the tables in order to align them e.g. the Table 1 says 245 (instead of 249) and Table 2 says 66* while the text in Line No. 220 is referring to sixty seven.

2. The presentation of the Table is little complicated as it is not clarified as to what is presented in the Table and what is not. In would be helpful for the reader to cite the Table or Figure at the first mention. The way the results are presented, the reader will only know where the table being referred to is when they reach the end of the paragraph.

3. In addition, navigating through the text and tables is a little tedious as the data is not presented according to the tables order. In the process, making locating what is being referred to in the text a little hard for the reader. For instance, Lines 192-197 appears to be referring to both Table 1 (asymptomatic participants) and Table 2(sensitivity). It would also be helpful for the reader if the authors can state when the results are "not shown" or are found in the supplementary.

Reviewer #3: This is a nice evaluation of seven RDTs for covid. However, diagnostic accuracy depends a lot on the study sample and participant mix. This is information that is lacking in this study and it is what I have most concerns about.

1. Participants were selected from travellers and people in isolation. To what extent are they representative for the population in which the RDTs will be used? For example, the conclusion says: "This RDT can be recommended for use among symptomatic patients in health facilities in order not to delay patient management while waiting for qRT-PCR results". But these are not the participants included.

2. For the reason mentioned in the previous comment, I do not think that this conclusion is valid. So please rephrase.

3. How were particpants selected? In a consecutive way? On particular days? This needs to be clarified.

4. How were the different tests allocated to the right person? At random, or in different time-periods? Please explain.

5. How many samples/tests per participant were included? Did some participants receive more than two tests? Please explain.

6. There are distinct differences in population mix (males/females; cases/controls) between the different tests. This could have influenced the accuracy, but the authors say little about these differences. Maybe this can be added?

7. In general, I think this manuscript would benefit from a better reporting. Please refer to the STARD guidelines for reporting of diagnostic accuracy studies.

8. RT-PCR is the reference standard, but very little information is provided. For example, I can imagine that having only one negative RT-PCR is not very reliable in determining a 'control' participant. Would it be possible to explian a bit more about the reference standard? And could the reference standard have led to bias? (which should then be mentioned in a limitations section in the Discussion)

9. The authors have analyzed the data against different Ct thresholds. As far as I can see, the Ct thresholds are different from what I have seen in other RDT analyses. Would it be possible for the authors to comment on these Ct thresholds? And how do they relate to clinical practice? Why were those Ct's chosen?

6. PLOS authors have the option to publish the peer review history of their article (what does this mean?). If published, this will include your full peer review and any attached files.

Reviewer #1: No

Reviewer #2: **Yes: **Ngonda Saasa

Reviewer #3: **Yes: **Mariska Leeflang

---

## [Author Response · Author response to Decision Letter 0]

5 Nov 2021

Response to Reviewers

PONE-D-21-24051

Field Evaluation of the Performance of Seven Antigen Rapid Diagnostic Tests for the Diagnosis of SARs-CoV-2 Virus Infection in Uganda

1. Information in the tables must be well presented and numbers must tally with those that are given in the text. The order of presentation of results in the text must follow the order in which they appear in the tables.

Response: 

The information in the tables and numbers in text have been reviewed and now tally. The order of the results in the text has been reorganised to follow the order in which they appear in the tables.

2. Provide more information about the participants in this study and how they were selected. Do they represent the population on which such the tests are intended to be used on? Was each test applied to every participant and if not, what impact would this have on the comparison of the test results.

Response

The participants were enrolled consecutively whenever RDT kits for evaluation were supplied to UVRI. The evaluation did not receive the RDTs at the same time and quantities supplied for the evaluation were different (depended on how many the suppliers would be able to supply). The majority of participants considered for the evaluation were those found admitted with symptoms known to be for COVID-19 patients. Since RDTs were not received at the same time and because of the urgency due to the pandemic, we were not able to evaluate the kits using the same participants as we would have wanted but rather to enrol those who consented to the evaluation. This is further complicated by the acute nature of the disease and rapid changes in viral loads in individuals. We observed that our sample had significantly more men than women and this was also observed in the centres used (more men in health facilities and border crossings). This means that if the RDTs are to be used at health facilities where individuals with COVID-19 like symptoms would go for health care, or at border crossings, then it is more likely that a similar profile of individuals will be the ones to be tested using recommended RDTs.

3. Provide more information about the validity of the RT-PCR that was used as a Gold Standard in this study.

Response

More information on the sensitivity and specificity of the RT-PCR that was used as the Gold Standard in this study has been included in the manuscript. 

Line162-166: The protocol has good sensitivity with a limit of detection (LoD) of 3.9 RNA copies per reaction for E-gene assay and 3.6 RNA copies per reaction for RdRp gene assay using in-vitro transcribed RNA identical to 2019 novel coronavirus sequences and specificity of 100%

Response

 Protocol used has been referenced. Not deposited in the protocols.io .

Response:

Plos One formatting requirements have been addressed

2.Please include additional information regarding the survey or questionnaire used in the study and ensure that you have provided sufficient details that others could replicate the analyses. For instance, if you developed a questionnaire as part of this study and it is not under a copyright more restrictive than CC-BY, please include a copy, in both the original language and English, as Supporting Information.

Response:

No questionnaire has been included. Because the survey form was in excel. A copy of the Excel dataset used for data collection and which was transformed to STATA for the analysis can be accessed using the link below;-

http://eaccr.org/sites/default/files/2021-11/Field%20Evaluation%20of%20the%20Performance%20of%20Seven%20Antigen%20Rapid%20Diagnostic%20Tests.xlsx

[RD is a consultant at Abbott. Therefore, he did not participate in the investigation of Panbio™ COVID-19 Ag Rapid test performance.]. 

Response

Updated conflict of interest statement: 

RD is a consultant at Abbott. Therefore he did not participate in the investigation of Panbio™ COVID-19 Ag Rapid test performance. This does not alter our adherence to PLOS ONE policies on sharing data and materials. 

Response: The minimum Dataset has been made available. It can be accessed using the link below;-

http://eaccr.org/sites/default/files/2021-11/Field%20Evaluation%20of%20the%20Performance%20of%20Seven%20Antigen%20Rapid%20Diagnostic%20Tests.xlsx

5. One of the noted authors is a group or consortium [EPI Laboratory team and UVRI COVID 19 Technical team.] In addition to naming the author group, please list the individual authors and affiliations within this group in the acknowledgments section of your manuscript. Please also indicate clearly a lead author for this group along with a contact email address.

Response:

Line 651-670: The members in EPI Laboratory team and UVRI COVID 19 Technical team have been named in the acknowledgement section. The author affiliation has been included and the Lead authors in the group named.

Response: The statement with this reference has been deleted from the manuscript. It was not a core part of the research being presented. 

Reviewers' comments:

Reviewer's Responses to Questions

Comments to the Author

1. Is the manuscript technically sound, and do the data support the conclusions?

Reviewer #1: Partly

Reviewer #2: Yes

Reviewer #3: Partly

2. Has the statistical analysis been performed appropriately and rigorously?

Reviewer #1: Yes

Reviewer #2: No

Reviewer #3: Yes

3. Have the authors made all data underlying the findings in their manuscript fully available?

Reviewer #1: No

Reviewer #2: Yes

Reviewer #3: No

Response: The data has been made available. It can be found using the following URL

http://eaccr.org/sites/default/files/2021-11/Field%20Evaluation%20of%20the%20Performance%20of%20Seven%20Antigen%20Rapid%20Diagnostic%20Tests.xlsx

4. Is the manuscript presented in an intelligible fashion and written in standard English?

Reviewer #1: Yes

Reviewer #2: Yes

Reviewer #3: Yes

5. Review Comments to the Author

Reviewer #1: The manuscript is interesting and well written.

The authors should acknowledge more limitations of the study. The comparison between the tests is not fair, because the tests are not compared in a randomized way. Sensitivity and specificity can be different if samples come from populations with different disease spectrum/prevalence.

The STARD checklist should be used (https://www.equator-network.org/reporting-guidelines/stard).

Response: 

Line 625-631 : Limitations why we could not use the mentioned randomization design have been added in the manuscript. Basically these kits were evaluated as and when the suppliers brought them to the UVRI. This was at different times and in different quantities. Because of the urgency to identify an RDT that would help in the diagnosis of COVID-19 we evaluated as soon as we received the kits and used participant sources established by the Ugandan Ministry of Health (Isolation centres and points of entry)

-We have followed the STARD checklist when reporting in the manuscript.

Table 1 should be completed with percentages for all the variables (e.g. age groups).

Response

Table has been completed with percentages for all the variables (age groups and the grouping according to duration of symptoms now have percentages).

Tables should be formatted in a better way.

Responses

Table 1, 2 and 3 have been formatted.

Is the time lag between Antigen Test and PCR a possible explanation for the difference observed with vendors diagnostic performance?

Response: 

There is a time lag between the Antigen test and PCR which range from 1 to 37 days with a median of 4-7 days except for RESPI Strip which had a median of 21 days. We think this did not affect the diagnostic performance of the kits since the samples were stored at -80oC during that time. Please see table below: This information has been included in the manuscript.

Table : Days between the Rapid Diagnostic Test and the PCR test

Ag RDT Median days Mean Min Max

Biocredit 4 7.3 1 21

Respi 21 22.7 1 37

PCL 7 9.7 1 24

MedSan 5 4.9 2 12

PanBio 4 3.7 2 7

Colloidal/Novegent 4 4.3 1 11

VivaDiag 5 7.5 2 28

Reviewer #2: Reviewer Recommendation and Comments for Manuscript Number PONE-D-21-24051

The comparison of the panel of tests is key to improving the use of these tests in local situation. Such is very important and carrying out such investigations improve the performance as mitigation measures can be put in place in real time. The fact that such tests can be performed at the bedside therefore hasten the recovery process and facilitates a speedy intervention and therefore recovery.

Observation

1. There is need for the authors to revisit the figures (number/values) in the text as well as those in the tables in order to align them e.g. the Table 1 says 245 (instead of 249) and Table 2 says 66* while the text in Line No. 220 is referring to sixty seven.

Response: We have aligned the figures in the tables and text.

2. The presentation of the Table is little complicated as it is not clarified as to what is presented in the Table and what is not. In would be helpful for the reader to cite the Table or Figure at the first mention. The way the results are presented, the reader will only know where the table being referred to is when they reach the end of the paragraph.

Response 

We have maintained reference to the table at the end since we did not report per table but per kit. Referring to the table at the beginning for each kit would make the presentation monotonous

3. In addition, navigating through the text and tables is a little tedious as the data is not presented according to the tables order. In the process, making locating what is being referred to in the text a little hard for the reader. For instance, Lines 192-197 appears to be referring to both Table 1 (asymptomatic participants) and Table 2(sensitivity). It would also be helpful for the reader if the authors can state when the results are "not shown" or are found in the supplementary.

Response

We have rearranged the text to follow the order of the tables presented for the specific kit being evaluated.

Reviewer #3: This is a nice evaluation of seven RDTs for covid. However, diagnostic accuracy depends a lot on the study sample and participant mix. This is information that is lacking in this study and it is what I have most concerns about.

1. Participants were selected from travellers and people in isolation. To what extent are they representative for the population in which the RDTs will be used? For example, the conclusion says: "This RDT can be recommended for use among symptomatic patients in health facilities in order not to delay patient management while waiting for qRT-PCR results". But these are not the participants included.

Response

The participants were enrolled consecutively whenever RDT kits for evaluation were supplied to UVRI. The evaluation did not receive the RDTs at the same time and quantities supplied for the evaluation were different (depended on how many the suppliers would be able to supply). The majority of participants considered for the evaluation were those found admitted with symptoms known to be for COVID-19 patients. Since RDTs were not received at the same time and because of the urgency due to the pandemic, we were not able to evaluate the kits using the same participants as we would have wanted but rather to enrol those who consented to the evaluation. This is further complicated by the acute nature of the disease and rapid changes in viral loads in individuals. We observed that our sample had significantly more men than women and this was also observed in the centres used (more men in health facilities and border crossings). This means that if the RDTs are to be used at health facilities where individuals with COVID-19 like symptoms would go for health care, or at border crossings, then it is more likely that a similar profile of individuals will be the ones to be tested using recommended RDTs.

In addition, the Isolation Centers had patients that were admitted from different parts of the country.

2. For the reason mentioned in the previous comment, I do not think that this conclusion is valid. So please rephrase.

Response

Line 593-604: The conclusion has been rephrased.

3. How were particapants selected? In a consecutive way? On particular days? This needs to be clarified.

Response 

Participants were recruited consecutively, and these were found at the sites of recruitment. Also recruitment for the different kits was as when the kits were received from the suppliers. 

We have clarified this in the manuscript.

4. How were the different tests allocated to the right person? At random, or in different time-periods? Please explain.

Response 

The different kits were allocated at different time periods as the different tests were received from the suppliers. 

The dates when the different participants for the different kits were recruited is mentioned in the manuscript

5. How many samples/tests per participant were included? Did some participants receive more than two tests? Please explain.

Most participants/samples had only one test. This was because of how we received the samples and also when samples were taken off, they were put in the buffer supplied with that kit. A few samples were used to evaluate more than one test however a comparison of the results has not been included in the manuscript. Additionally, one RDT (RESPI) did not have a buffer but shared a sample with real time PCR. 

Seventy- six samples had both PCL COVID19 Ag Rapid FIA and COVID-19 Ag Respi-Strip. Fifteen samples had both BIOCREDIT COVID -19 Ag Test and COVID-19 Ag Respi-Strip used. As mentioned above, no comparison between results from these tests has been included in the manuscript.

Line 332-335: The information about numbers of samples used to evaluate more than one RDT has been included in the manuscript.

6. There are distinct differences in population mix (males/females; cases/controls) between the different tests. This could have influenced the accuracy, but the authors say little about these differences. Maybe this can be added?

Response

The population mix (cases/controls ) between the different tests has been added as a study limitation.

However, we do not think the difference in male/female mix could have affected the accuracy of the different tests.

Response

7. In general, I think this manuscript would benefit from a better reporting. Please refer to the STARD guidelines for reporting of diagnostic accuracy studies.

Response

We have referred to the STARD guidelines for reporting diagnostic accuracy studies and have made some edits to the manuscripts.

8. RT-PCR is the reference standard, but very little information is provided. For example, I can imagine that having only one negative RT-PCR is not very reliable in determining a 'control' participant. Would it be possible to explian a bit more about the reference standard? And could the reference standard have led to bias? (which should then be mentioned in a limitations section in the Discussion)

Response

Line 162-166: We have added more information concerning the RT-PCR we used as the reference standard: This assay was selected as the Standard reference for the kit evaluation because the protocol has good sensitivity with a limit of detection (LoD) of 3.9 RNA copies per reaction for E-gene assay and 3.6 RNA copies per reaction for RdRp gene assay using invitro transcribed RNA identical to 2019 novel coronavirus sequences and specificity of 100%. It was also recommended by WHO. 

Since the specificity of the test was 100%, we believe using only one negative RT-PCR result for the control participants did not cause any bias in our study.

9. The authors have analyzed the data against different Ct thresholds. As far as I can see, the Ct thresholds are different from what I have seen in other RDT analyses. Would it be possible for the authors to comment on these Ct thresholds? And how do they relate to clinical practice? Why were those Ct's chosen?

Response

The authors chose the Ct-thresholds in reference to literature and after discussions in-country.

A Ct ≤ 39: This is the cut off for positivity provided by the Charite Berlin PCR Protocol that we used in the analysis (Corman VM et al., 2020).

A Ct≤29 was used because literature showed that patients with Ct value below 30 were found to be more infectious due to the high viral load and these Ct values were more likely to be found in the first days of the infection. Since we are looking for kits that will detect the infection early we decided to include this cut off in our analysis.

https://www.sciencemag.org/news/2020/09/one-number-could-help-reveal-how-infectious-covid-19-patient-should-test-results

(Bayat SA et al. 2021) 

A Ct value of ≤ 33 for the E-gene were found not to be contagious. No viruses were cultured from the patients with Ct value greater than 34 in a study carried out in France. Therefore, these were taken to be of less clinical value (La Scola B et al. 2020). Since we still need to identify these patients that is why we had this cut off value although our final report is on RdRp gene Ct value.

6. PLOS authors have the option to publish the peer review history of their article (what does this mean?). If published, this will include your full peer review and any attached files.

Do you want your identity to be public for this peer review? For information about this choice, including consent withdrawal, please see our Privacy Policy.

Reviewer #1: No

Reviewer #2: Yes: Ngonda Saasa

Reviewer #3: Yes: Mariska Leeflang

---

## [Decision Letter · Decision Letter 1]

22 Nov 2021

PONE-D-21-24051R1Field Evaluation of the Performance of Seven Antigen Rapid Diagnostic Tests for the Diagnosis of SARs-CoV-2 Virus Infection in UgandaPLOS ONE

Dear Dr. Bwogi,

Thank you for submitting your manuscript to PLOS ONE. After careful consideration, we feel that it has merit but does not fully meet PLOS ONE’s publication criteria as it currently stands. Therefore, we invite you to submit a revised version of the manuscript that addresses the points raised during the review process.

The reviewers are of the view that you did not adequately atted to their concerns. Please pay partcular attention to those raised by reviewer No. 3. Attend to all the concerns raised and return a revised manuscript as advised in this letter.

We look forward to receiving your revised manuscript.

Kind regards,

Martin Chtolongo Simuunza, PhD

Academic Editor

PLOS ONE

Journal Requirements:

Reviewers' comments:

Reviewer's Responses to Questions

**Comments to the Author**

1. If the authors have adequately addressed your comments raised in a previous round of review and you feel that this manuscript is now acceptable for publication, you may indicate that here to bypass the “Comments to the Author” section, enter your conflict of interest statement in the “Confidential to Editor” section, and submit your "Accept" recommendation.

Reviewer #1: (No Response)

Reviewer #2: All comments have been addressed

Reviewer #3: (No Response)

2. Is the manuscript technically sound, and do the data support the conclusions?

Reviewer #1: Yes

Reviewer #2: Yes

Reviewer #3: Partly

3. Has the statistical analysis been performed appropriately and rigorously? 

Reviewer #1: Yes

Reviewer #2: Yes

Reviewer #3: Yes

4. Have the authors made all data underlying the findings in their manuscript fully available?

Reviewer #1: Yes

Reviewer #2: Yes

Reviewer #3: Yes

5. Is the manuscript presented in an intelligible fashion and written in standard English?

Reviewer #1: Yes

Reviewer #2: Yes

Reviewer #3: No

6. Review Comments to the Author

Reviewer #1: Thanks for having addressed all my comments. I think that the manuscript is improved but there is still one thing missed. STARD checklist should be uploaded as supplementary material and the authors should explicitly state in what page/section the checklist item has been covered.

https://www.equator-network.org/wp-content/uploads/2015/10/STARD-2015-Checklist.docx

Reviewer #2: The authors have adequately addressed the concerns. The tables have beed realigned and some of the inconsistencies in the numbers in the table and text have been attended to.

Reviewer #3: The authors have improved their manuscript. However, there are a few remaining comments that have not or insufficiently have been addressed.

1. The selection criteria have not been clearly described. The authors now describe that participants were enrolled consecutively, but it is not clear in what setting the study was done or whether the patients enrolled in this study were representative. Also, the authors write that they included cases and controls, but that is ambiguous language. Please check the STARD guidelines about what to report about the selection criteria and the selection process.

2. The authors state that they followed the STARD guidelines, but there is still information missing. Please check. Also, the STARD guidelines recommend a flow chart to describe visually how patients enroll in the study and the tests they undergo and what the test results are. Such a flow chart would be really helpful in this case.

3. From the response to my comment about the timing of the tests, I understand that every patient is tested by only one of the index tests (and the reference standard). So patients enrolled at the start of the study will have received a different test than patients enrolled at the end of the study. That makes a comparisons between tests a bit difficult. So actually, authors should have mentioned this as a limitation. (see also my previous comment 6)

4. The authors provided now more information about the reference standard, but not how it was really used to make a diagnosis (e.g. for a participant who does not have COVID-19, should the test have been negative twice?).

7. PLOS authors have the option to publish the peer review history of their article (what does this mean?). If published, this will include your full peer review and any attached files.

Reviewer #1: No

Reviewer #2: No

Reviewer #3: **Yes: **Mariska Leeflang

---

## [Author Response · Author response to Decision Letter 1]

20 Jan 2022

Responses to Reviewers

PONE-D-21-24051R1

Field Evaluation of the Performance of Seven Antigen Rapid Diagnostic Tests for the Diagnosis of SARs-CoV-2 Virus Infection in Uganda

Response:

References have been reviewed and changes have been made to reference No 6 and 7.

Reference no 6. Bulilete O, Lorente P, Leiva A, Carandell E, Oliver A, Rojo E, et al. Evaluation of the Panbio rapid antigen test for SARS-CoV-2 in primary health care centers and test sites. medRxiv. 2020.

Replaced with more up to date reference below: 

Bulilete O, Lorente P, Leiva A, Carandell E, Oliver A, Rojo E, Pericas P, Llobera J, COVID-19 Primary Care Research Group. Panbio™ rapid antigen test for SARS-CoV-2 has acceptable accuracy in symptomatic patients in primary health care. Journal of Infection. 2021 Mar 1;82(3):391-8.

Reference no 7.Olearo F, Noerz D, Heinrich F, Sutter JP, Roedel K, Schultze A, et al. Handling and accuracy of four rapid antigen tests for the diagnosis of SARS-CoV-2 compared to RT-qPCR. medRxiv. 2020.

Replaced with the reference below: The reference above is no longer on the website. The reference available is the one below

Olearo F, Nörz D, Heinrich F, Sutter JP, Roedl K, Schultze A, Zur Wiesch JS, Braun P, Oestereich L, Kreuels B, Wichmann D. Handling and accuracy of four rapid antigen tests for the diagnosis of SARS-CoV-2 compared to RT-qPCR. Journal of Clinical Virology. 2021 Apr 1;137:104782.

2. Thanks for having addressed all my comments. I think that the manuscript is improved but there is still one thing missed. STARD checklist should be uploaded as supplementary material and the authors should explicitly state in what page/section the checklist item has been covered.

Response

The STARD checklist has been attached to the rebuttal letter. In the checklist we have indicated the line no where the checklist has been covered. We have not included the STARD checklist in the supplementary material since we could not reference it in the manuscript

3. The selection criteria have not been clearly described. The authors now describe that participants were enrolled consecutively, but it is not clear in what setting the study was done or whether the patients enrolled in this study were representative. Also, the authors write that they included cases and controls, but that is ambiguous language. Please check the STARD guidelines about what to report about the selection criteria and the selection process.

Response

Line 84-93. More information has been included in the section of the selection criteria and process. 

Line 27-28. Cases and Controls have been replaced with PCR positive and PCR negative participants.

4. The authors state that they followed the STARD guidelines, but there is still information missing. Please check. Also, the STARD guidelines recommend a flow chart to describe visually how patients enroll in the study and the tests they undergo and what the test results are. Such a flow chart would be really helpful in this case.

Response

• We have reviewed the STARD guidelines to guide us on inclusion of missing information in the manuscript. The STARD checklist has been attached to indicate where the different sections of the STARD checklist are in the paper.

• Flow charts showing patient enrollment and tests have been placed as figures in the supplementary material.

5. From the response to my comment about the timing of the tests, I understand that every patient is tested by only one of the index tests (and the reference standard). So patients enrolled at the start of the study will have received a different test than patients enrolled at the end of the study. That makes a comparisons between tests a bit difficult. So actually, authors should have mentioned this as a limitation. (see also my previous comment 6)

Response

Line 491-493. “Difficult in comparison of the tests because of timing of the tests and thus every patient tested with one of the index tests” has been included in the study limitation.

6. The authors provided now more information about the reference standard, but not how it was really used to make a diagnosis (e.g. for a participant who does not have COVID-19, should the test have been negative twice?).

Response : Line 137-146: 

This has now been described: A sample was considered positive if it had a Ct value ≤39 on the confirmatory qRT-PCR. A sample was considered negative if the Ct value was undetermined on the screening qRT-PCR or if it was positive on the screening qRT-PCR but undetermined or >39 on the confirmatory qRT-PCR.

---

## [Decision Letter · Decision Letter 2]

1 Mar 2022

Field Evaluation of the Performance of Seven Antigen Rapid Diagnostic Tests for the Diagnosis of SARs-CoV-2 Virus Infection in Uganda

PONE-D-21-24051R2

Dear Dr. Bwogi,

We’re pleased to inform you that your manuscript has been judged scientifically suitable for publication and will be formally accepted for publication once it meets all outstanding technical requirements.

Kind regards,

Martin Chtolongo Simuunza, PhD

Academic Editor

PLOS ONE

Additional Editor Comments (optional):

Reviewers' comments:

Reviewer's Responses to Questions

**Comments to the Author**

1. If the authors have adequately addressed your comments raised in a previous round of review and you feel that this manuscript is now acceptable for publication, you may indicate that here to bypass the “Comments to the Author” section, enter your conflict of interest statement in the “Confidential to Editor” section, and submit your "Accept" recommendation.

Reviewer #1: All comments have been addressed

Reviewer #3: All comments have been addressed

2. Is the manuscript technically sound, and do the data support the conclusions?

Reviewer #1: Yes

Reviewer #3: Yes

3. Has the statistical analysis been performed appropriately and rigorously? 

Reviewer #1: Yes

Reviewer #3: Yes

4. Have the authors made all data underlying the findings in their manuscript fully available?

Reviewer #1: Yes

Reviewer #3: Yes

5. Is the manuscript presented in an intelligible fashion and written in standard English?

Reviewer #1: Yes

Reviewer #3: Yes

6. Review Comments to the Author

Reviewer #1: (No Response)

Reviewer #3: I have no further comments, because all my previous comments have been addressed in one way or another.

7. PLOS authors have the option to publish the peer review history of their article (what does this mean?). If published, this will include your full peer review and any attached files.

Reviewer #1: No

Reviewer #3: **Yes: **Mariska M.G. Leeflang

---

## [Editor Report · Acceptance letter]

23 Mar 2022

PONE-D-21-24051R2 

Field Evaluation of the Performance of Seven Antigen Rapid Diagnostic Tests for the Diagnosis of SARs-CoV-2 Virus Infection in Uganda 

Dear Dr. Bwogi:

I'm pleased to inform you that your manuscript has been deemed suitable for publication in PLOS ONE. Congratulations! Your manuscript is now with our production department. 

Kind regards, 

on behalf of

Dr. Martin Chtolongo Simuunza 

Academic Editor

PLOS ONE